# Synthesis and Characterization of Benzoxazine Resin Based on Furfurylamine

**DOI:** 10.3390/ma15238364

**Published:** 2022-11-24

**Authors:** Jing Wang, Qinghua Liu, Jiangnan Yu, Riwei Xu, Chengzhong Wang, Jinping Xiong

**Affiliations:** 1Basic Education School, Beijing Information Technology College, Beijing 100070, China; 2Beijing Key Laboratory of Electrochemical Process and Technology for Materials, Beijing University of Chemical Technology, Beijing 100029, China

**Keywords:** furfurylamine, benzoxazine, bisphenol A-furfuryl amine oxazine, POSS

## Abstract

This paper presents an investigation of the modification of natural oxazines to traditional bisphenol A benzoxazines. Eugenol was reacted with furfurylamine to synthesize a new type of benzoxazine (eugenol–furfurylamine benzoxazine), with a yield of 77.65%; and another new type of benzoxazine (bisphenol A–furfurylamine benzoxazine) was generated from bisphenol A and furfurylamine, with the highest yield of 93.78%. In order to analyze and study the target molecules, IR (infrared radiation) spectroscopy, GPC (gel-permeation chromatograph), mass spectrometry, 1H-NMR (nuclear magnetic resonance), DSC (differential scanning calorimetry), and DMA (dynamic mechanical analysis) tests were conducted. Eugenol-furfurylamine benzoxazine and conventional bisphenol A-aniline benzoxazine (BZ) composite was also analyzed and cured at different mass ratios of 2:98, 5:95, 10:90, 20:80, and 40:60. When the content of eugenol furfurylamine in the blend reached 5%, the strength of the composite was greatly enhanced, while the strength decreased with the increase in eugenol furfurylamine oxazine content. Moreover, octamaleimide phenyl POSS (OMPS, polyhedral oligomeric silsesquioxane) and bisphenol A furamine benzoxazine were mixed at different molar ratios of 1:16, 1:8, 1:4, 1:2, and 1:1. The curing temperature sharply decreased with the increase in OMPS content. When the molar ratio reached 1:1, the curing temperature decreased from 248 to 175℃. A further advantage of using eugenol and furfurylamine is that they are renewable resources, which is important in terms of utilizing resources effectively and developing environmentally friendly products.

## 1. Introduction

For more than 100 years, phenolic resins have been always used in commercial applications. Because of their low cost and dimensional stability, they continue to be widely used in industry. Based on traditional phenolic resin, benzoxazine resin becomes a new thermosetting resin that contains a six-membered heterocyclic carbon and nitrogen chain. The potential of this new engineering plastic has attracted the attention of scholars [1,2]. In addition to its excellent properties (such as high heat resistance, flame retardancy, excellent electrical and chemical properties, low water absorption, and low cost), traditional phenolic resin does not emit small molecules during the curing process, and its curing shrinkage and porosity are low. Moreover, polybenzoxazine has special advantages such as a low coefficient of thermal expansion, good high-temperature resistance, mechanical properties, and good moisture resistance. However, in the study of new thermosetting resins, it is found that monofunctional benzoxazine has a chain transfer reaction during ring opening polymerization, and the molecular weight of the benzoxazine polymer obtained is relatively low, which limits its application scope. Although bifunctional benzoxazines (such as bisphenol A type) can be used as high-performance materials, due to the characteristics of their molecular structures, such benzoxazines have the disadvantages of low crosslinking density, brittleness, and poor toughness. Therefore, in order to meet the requirements of special use, benzoxazine needs to be properly modified [3,4,5], and the modification of benzoxazine with epoxy resin has therefore been extensively studied [6].

In a study by Kimura et al. [7], benzoxazine was used as a curing agent for epoxy resin preparation. Curing the accelerator was unnecessary, and the epoxy resin modified by benzoxazine resin exhibited good thermal stability below 150℃, while the curing reaction proceeded rapidly above 150℃. In comparison with the products cured with phenolic resin, products cured with bisphenol A type benzoxazine were more heat-resistant, water-resistant, electrically insulated, and mechanically efficient.

Ishida et al. [8] produced the copolymers of bisphenol A epoxy resin and bisphenol A benzoxazine without a catalyst. An analysis of the dynamic mechanical thermal properties of the copolymer revealed that the crosslink density was higher than that of benzoxazine. In an imide reaction between octamethylphenyl caged sesquisiloxane and maleic anhydride, octahydridrosilsesquioxane-containing bismaleimide resins could be synthesized. In comparison with homopolymers, these copolymers possessed stronger mechanical properties, a higher glass-transition temperature (Tg), and better thermal stability [9].

Jun Zhang et al. [10] at Beijing University of Chemical Technology investigated a novel polyhedral oligomeric silsesquioxane (BZ-POSS) containing polybenzoxazine groups. As a result of the reaction of octanamine-based silsesquioxane with p-hydroxycresol and paraformaldehyde, each BZ-POSS molecule contained around 7.6 benzoxazine groups. When BZ-POSS was molten, it was miscible with bisphenol A-type benzoxazine (BZ).

In this area, Lee et al. [11,12] also produced a high-performance nanocomposite through the copolymerization of vinyl-containing benzoxazine with POSS. As a result, when 5% POSS was added to the copolymer, the Tg was increased from 307 to 333 °C.

In this paper, the benzoxazine monomer was designed to synthesize a new structure of benzoxazine based on furfuramine, which effectively improved the comprehensive performance of its cured products. On this basis, traditional bisphenol A oxazine was modified to improve its mechanical properties. The curing temperature of the prepared bisphenol A furylamine benzoxazine was lowered by adding octamaleimide phenyl POSS (OMPS). The raw materials of furfurylamine and eugenol are all from natural products and have wide sources, low pollution, and a low price. This is of great significance for sustainable development and the development of environmentally friendly materials.

## 2. Experiment

### 2.1. Material Preparation

#### 2.1.1. Preparation of Eugenol–Furfurylamine Benzoxazine

In total, 200 mL toluene, 20.5 g (0.125 mol) of eugenol, and 21.5 g (0.125 mol) of furfurylamine were added in turn to a 500 mL three-necked flask with a condenser and stirring. Then, 30 g (0.5 mol) paraformaldehyde was gradually added under strong stirring in ice water. At room temperature, the mixture was vigorously stirred and then placed in an oil bath at 95 °C for two hours. Upon completion of the reaction, 1 mol/L of sodium hydroxide was added into and stirred; then, the mixture was poured into a separating funnel to separate the product, which was washed with sodium hydroxide and neutralized with deionized water. After the product was poured into a porcelain tray and dried under vacuum at 60 °C for 24 h, a reddish-brown solid with a yield of 77.65% was obtained.

#### 2.1.2. Preparation of Bisphenol A-Furfurylamine Benzoxazine

In total, 200 mL toluene, 71.5 g (0.25 mol) of bisphenol A, and 21.5 g (0.125 mol) of furfurylamine were added in succession to a 500 mL three-neck flask with condenser and stirring, and 30 g (0.5 mol) paraformaldehyde was subsequently added in an ice bath under strong stirring. After stirring vigorously at room temperature, the mixture was placed in an oil bath at 95 °C for two hours. In order to separate the product, 1 mol/L of sodium hydroxide was added and stirred to mix evenly; then, the mixture was poured into a separator funnel to separate the product, which was washed with sodium hydroxide and then neutralized with deionized water. After the product was poured into a porcelain tray and dried under vacuum at 60 °C for 24 h, a reddish-brown solid with a highest yield of 74.23% was obtained.

#### 2.1.3. The Co-Curing of Eugenol–Furfurylamine/BZ

Eugenol–furfurylamine oxazines were co-cured with traditional BZ at a mass ratio of 2:98. Via the solution-mixing method, the BZ and eugenol–furfurylamine oxazines were dissolved in a certain amount of tetrahydrofuran and stirred for five hours before the solvent was removed in vacuo. A certain amount of the above blended system was placed in a mold and cured in a vacuum oven. The curing processes were as follows: 120 °C/2 h, 140 °C/2 h, 160 °C/2 h, 180 °C/3 h, and 200 °C/2 h.

#### 2.1.4. OMPS/Bisphenol A–Furfurylamine Benzoxazine Blend

The synthesis of OMPS was described in the references [13,14]. The composition formula can been seen in Figure 1.

Benzoxazine resin prepared with OMPS/bisphenol A–furfurylamine; the composition formula is shown in Figure 2.

A mixture of OMPS and bisphenol A–furfurylamine benzoxazine resin was dissolved in acetone at different molar ratios of 1:1, 1:2, 1:4, 1:8, and 1:16, at room temperature, and the solvent was evaporated in a vacuum.

### 2.2. Characterization

The products were tested by infrared spectroscopy (FTIR) (Nicolet-60 SXB Fourier infrared spectrometer; KBr tableting, scanning range: 400–4000 cm^−1^, Zequan International Group Shanghai Zequan Instrument Equipment Co., Ltd., Shanghai, China); gel permeation chromatography (GPC) (GPC 515-2410 system, Jinan Saichang Scientific Instrument Co., Ltd., Jinan, China); mass spectrometry (MS) (Waters Quattro Premier XE tandem quadrupole mass spectrometer; cone hole voltage: 30 V; Hubei Jiedao Scientific Instrument Co., Ltd., Hubei Province, China); the nuclear magnetic resonance test (^1^H-NMR) (the Bruker Avance 600 MHZ nuclear magnetic resonance instrument; CDCl_3_ as a solvent, Si (CH_3_) _4_ as Internal Reference Standards for Quantitative NMR, tested in room temperature; Shanghai Shidande Standard Technical Service Co., Ltd., Shanghai, China); the DSC test (the Perkin Elmer Pyris 1 thermal analyzer; an N_2_ environment, at a temperature rising rate of 10 °C/min, with Al_2_O_3_ as a reference material; the test temperature range: room temperature to 350 °C; Perkin Elmer Enterprise Management (Shanghai) Co., Ltd., Shanghai, China). The dynamic mechanical analysis (DMA) test (DMA 242 E, NETZSCH Scientific Instrument Trading (Shanghai) Co., Ltd., Shanghai, China) was used to characterize the relationship between the dynamic thermo-mechanical properties of the product and the frequency and temperature.

## 3. Results and Discussion

### 3.1. Determination of Synthesis Conditions

#### Eugenol–Furfurylamine Benzoxazine

(1) Table 1 showed the yield rates of eugenol–furfurylamine oxazine under different solvent conditions (toluene, chloroform, n-butanol, and DMF). As long as the dielectric constant of the solvent exceeded 4, no oxazine was generated in the product, so toluene was selected as the solvent.

(2) While the other conditions remained the same, the aldehyde species were changed (0.2 mol): the yields of paraformaldehyde, trioxane, formaldehyde aqueous solution, and eugenol–furfurylamine oxazine are shown in Table 2. When paraformaldehyde was used as a reactant, paraformaldehyde was chosen as the reactant of choice due to the high yield of the product.

(3) The other reaction conditions remained unchanged, and the amount of aldehyde was changed: (paraformaldehyde) 0.2 mol, 0.3 mol, 0.4 mol, 0.5 mol, and 0.6 mol. Table 3 showed the yield of eugenol–furfurylamine oxazine. In general, the greater the amount of paraformaldehyde used, the higher the yield. However, the yield remained at a certain level, which can been seen directly in Figure 3. Therefore, the chosen amount of paraformaldehyde was 0.4 mol.

### 3.2. Testing and Characterization of Eugenol–Furfurylamine Benzoxazine

Eugenol–furfurylamine oxazine’s structure was represented as Figure 4. In order to verify the success of the experimental synthesis of this benzoxazine, a series of tests and characterizations were conducted.

The FTIR spectrum of benzoxazine (Eugenol) was shown in Figure 5. The results were shown in Table 4.

There was no apparent absorption peak for the stretching vibration of the hydroxyl group that was detected near 3400 cm^−^^1^, indicating that the phenolic hydroxyl group had been completely reacted [16].

GPC and mass spectrometry were used to determine the molecular weight of the product. A GPC spectrum of benzoxazine (eugenol) was shown in Figure 6, from which it was evident that its molecular weight distribution was narrow, centered around 232 [17]. The mass spectra of benzoxazine (eugenol) was shown in Figure 7. According to Figure 7, benzoxazine (eugenol) had a molecular weight of 286 (the number circled in red), which was consistent with its theoretical molecular weight of 285.1 [18]. 

Figure 8 showed the ^1^H-NMR spectrum of benzoxazine (eugenol). The proton peaks in the figure: δ = 6.20–6.58 (a–e) was the proton peak on the benzene ring; the multiple peaks at 5.10 (k), 5.92 (j), and 3.30 (i) corresponded to the three kinds of protons a, b, and c, on CH_2_^a^=CH^b^-CH_2_^c,^ respectively; 4.87 (h) was the proton peak on -O-CH_2_-N-; 3.94 (g–f) was the proton peak on -Ar-CH_2_-N-; and 3.87 (l) was the proton peak on -O-CH_3_.

Peak area ratio: a:b:c:d:e:f:g:h:i:j:k:l = 1.10:1.00:0.94:1.13:1.09:2.20:1.99:2.09:2.21:0.92:1.92:1.95

Theoretical ratio: a:b:c:d:e:f:g:h:i:j:k:l = 1:1:1:1:2:2:2:2

It can be seen from the above data that the chemical shifts of hydrogen atoms at various positions of the product can infer the positions of hydrogen atoms in the product, and the peak area ratio was basically consistent with the theoretical value ratio. It can be inferred that the final product of the reaction was the designed target product.

In Figure 9, it can be seen that benzoxazine (eugenol) had a melting point of 84.8 °C and a broad heat absorption peaked near 230 °C on its DSC curing curve. There was no apparent curing exothermic peak in the entire DSC spectrum, and the monomer decomposed after 250 °C. As a result, it was presumed that monomeric cures were not possible for the subclass of pure oxazine [19]. A pure monomer of oxazine was incapable of self-curing.

### 3.3. Testing and Characterization of Bisphenol A–Furfurylamine Benzoxazine

Figure 10 was the structure of bisphenol A–furfurylamine benzoxazine. An extensive series of tests and characterizations of this benzoxazine were conducted in order to further verify its successful experimental synthesis.

The FTIR spectrum of benzoxazine (bisphenol A) was shown in Figure 11. The result was shown in Table 5.

GPC and mass spectrometry were used to determine the molecular weight of the product. As shown in Figure 12, the molecular weight distribution of benzoxazine (bisphenol A) was narrow, concentrated around 300–500. As shown in Figure 13, the molecular weight of the product was 471, while its theoretical molecular weight was 471, indicating that this was the target product [20]. 

Figure 14 showed the ^1^H-NMR spectrum of benzoxazine (bisphenol A). The proton peak in the figure δ = 6.252–6.987 (a–f) was the proton peak on the benzene ring; 4.85 (g) was the proton peak on -O-CH_2_-N-; 3.94 (i,h) was the proton peak on -Ar-CH_2_-N-; and 1.60 (j) was the proton peak on -O-CH_3_.

Peak area ratio: a:b:c:d:e:f:g:h:i:j = 1.21:1.20:1.03:1.24:1.00:1.03:2.05:1.99:2.14:3.06

Theoretical ratio: a:b:c:d:e:f:g:h:i:j = 1:1:1:1:1:1:2:2:2:3

It can be seen from the above data that the chemical shifts of hydrogen atoms at various positions of the product can infer the positions of hydrogen atoms in the product, and the peak area ratio was basically consistent with the theoretical value ratio. It can be inferred that the final product of the reaction was the designed target product.

### 3.4. Testing and Characterization of Eugenol–Furfurylaminobenzoxazine/BZ (Bisphenol A-Anilinooxazine)

Dynamic mechanical analysis (DMA) tests were conducted on the eugenol–furfurylamine benzoxazine/BZ-modified sample strips, and the results were shown in Figure 15. When eugenol–furfurylamine benzoxazine was added to BZ, the temperature region became narrow; the peak moved towards the higher temperature, and the loss factor increased [21]. Figure 16 showed the relationship between the energy-storage modulus and the temperature for BZ samples containing various levels of eugenol–furfurylamine oxazine: when the concentration reached 5%, the modulus greatly increased, indicating that the strength of the material was greatly enhanced. However, E’ decreased with an increase in eugenol–furfurylamine oxazine content.

### 3.5. Testing and Characterization of OMPS/Bisphenol A–Furfurylamine Benzoxazine

Figure 17 illustrated the IR spectra of OMPS and BPA at 1:1. Based on the comparison, it was evident that DA20 (the Diels–Alder reaction at 20 °C) and DA140 (the reverse Diels–Alder reaction at 140 °C) had identical infrared absorption peaks. There was evidence that a reversible reaction occurred between 20 and 100 °C [22].

A new peak was observed in the spectrum of DA100 at 1257 cm^−1^ (probably a new peak of C-O-C). By comparing DA300 with DA20, it was evident that the hydroxyl characteristic peaked at 3363 cm^−1^ was significantly larger, whereas the characteristic peaks of BZ monomered at 936 and 1229 cm^−1^ disappeared, while the characteristic peaked at 1483 cm^−1^ represented trisubstituted benzene. There was a characteristic peak at 1777 cm^−1^, which indicates that the DA reaction may have occurred at room temperature [23].

Based on Figure 18, we were able to determine that the curing temperature of pure bisphenol A–furfurylamine oxazine was significantly reduced when different proportions of OMPS were added, and the reduction increased with increasing OMPS content. Therefore, when the molar ratio of OMPS to bisphenol A–furfurylamine oxazine reached 1:1, the curing temperature was reduced from 248 to 175 °C.

## 4. Conclusions

The present study examined the synthesis of new types of benzoxazines from natural raw materials of eugenol and bisphenol A with furfurylamine, respectively. The IR, GPC, mass spectrometry, and ^1^H-NMR of the synthesized products confirmed that they were the desired benzoxazines, and the DSC and DMA test analyzed the performance of the target products. It was found that the mechanical properties of the materials were improved by adding eugenol–furfurylamine benzoxazine products to BZ. The strength of eugenol furylamine benzoxazine/BZ blends increased to maximum (when the eugenol furylamine benzoxazine content reached 5%) and then decreased with the increase in eugenol furylamine benzoxazine content. Additionally, as OMPS was added to bisphenol A–furfurylamine type oxazine, the curing temperature of the mixture reduced from 248 to 175 °C, and the higher the content of OMPS was, the greater the reduction became. The prepared new benzoxazine copolymer not only has stronger mechanical properties and better thermal stability than homopolymer but also comes from renewable resources, which is beneficial to the effective utilization of resources and the development of environmentally friendly materials.

## Figures and Tables

**Figure 1 materials-15-08364-f001:**
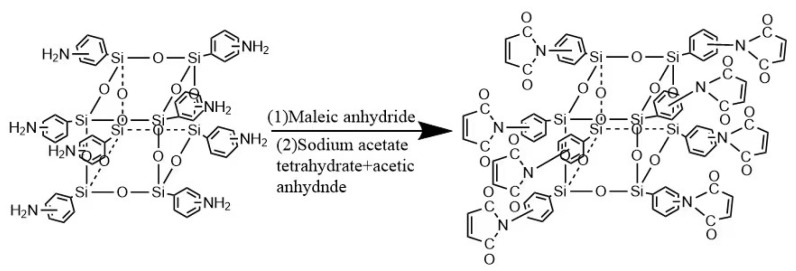
The reaction of OMPS.

**Figure 2 materials-15-08364-f002:**
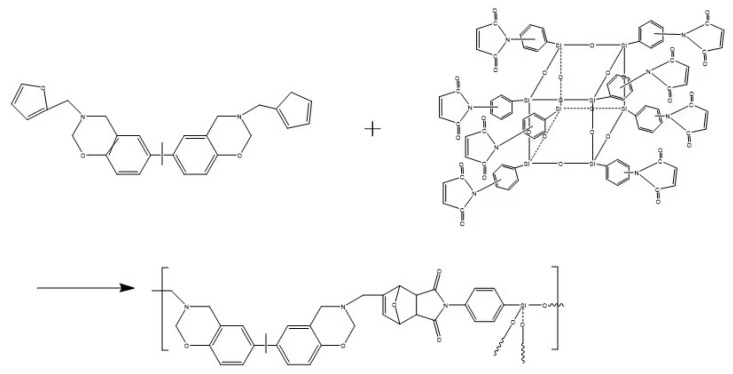
The reaction of OMPS and bisphenol A–furfuryl amine benzoxazine.

**Figure 3 materials-15-08364-f003:**
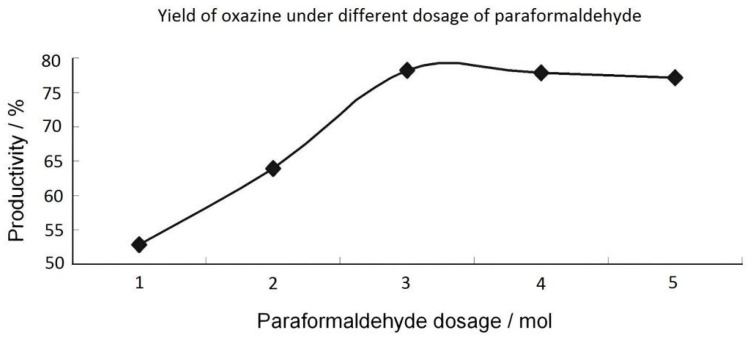
The product yield with the quantity of aldehydes.

**Figure 4 materials-15-08364-f004:**
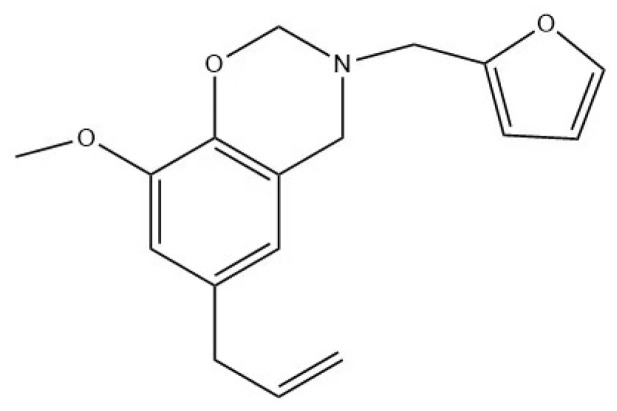
The structure of eugenol–furfurylamine benzoxazine.

**Figure 5 materials-15-08364-f005:**
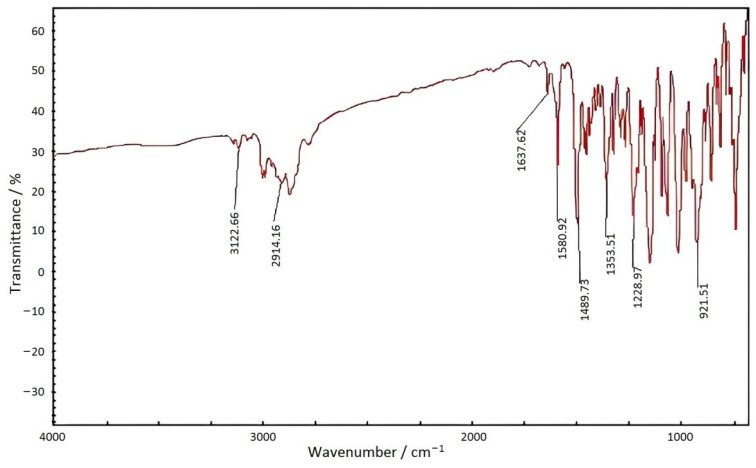
FTIR spectrum of benzoxazine (eugenol).

**Figure 6 materials-15-08364-f006:**
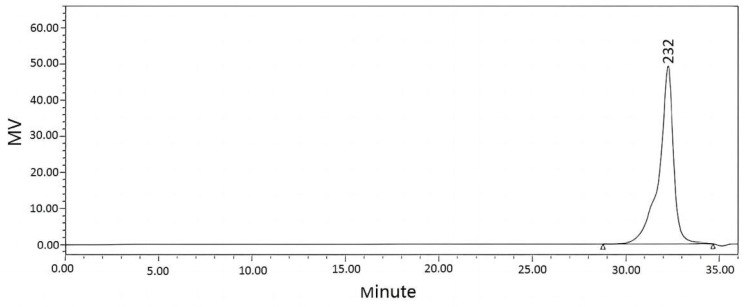
GPC of benzoxazine (eugenol).

**Figure 7 materials-15-08364-f007:**
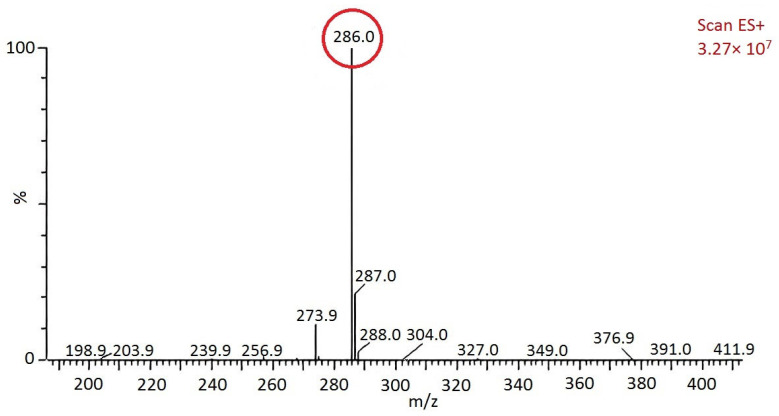
Mass spectrogram of benzoxazine (eugenol).

**Figure 8 materials-15-08364-f008:**
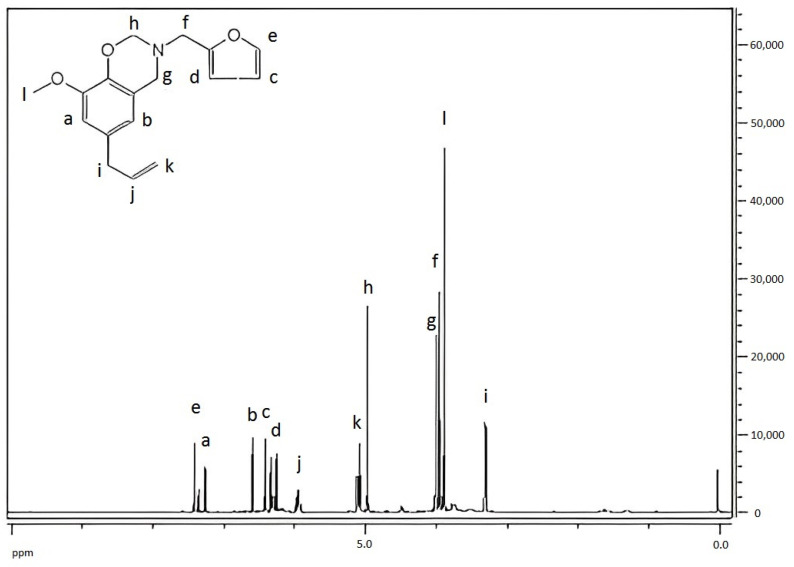
^1^H-NMR spectrum of benzoxazine (eugenol).

**Figure 9 materials-15-08364-f009:**
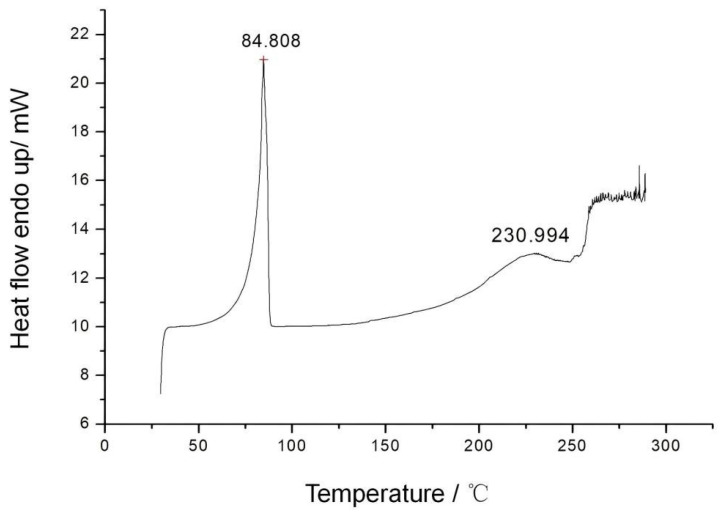
DSC curve of benzoxazine (eugenol).

**Figure 10 materials-15-08364-f010:**
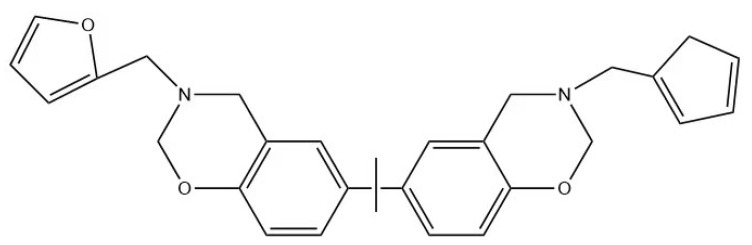
The structure of bisphenol-A-furfuryl amine benzoxazine.

**Figure 11 materials-15-08364-f011:**
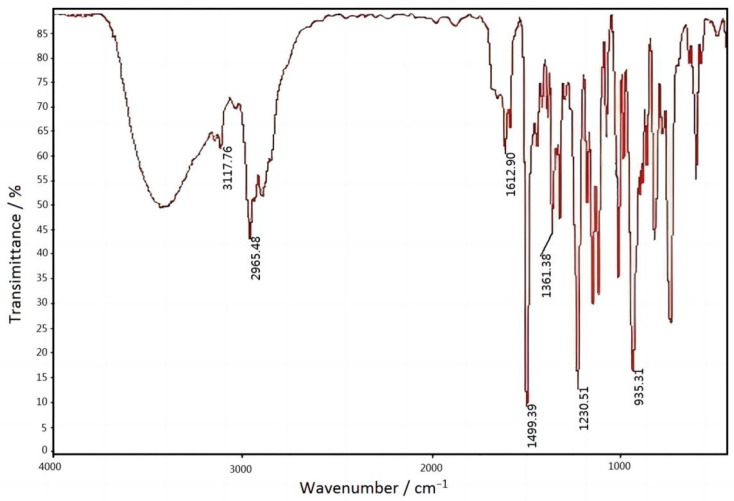
FTIR spectrum of benzoxazine (bisphenol A).

**Figure 12 materials-15-08364-f012:**
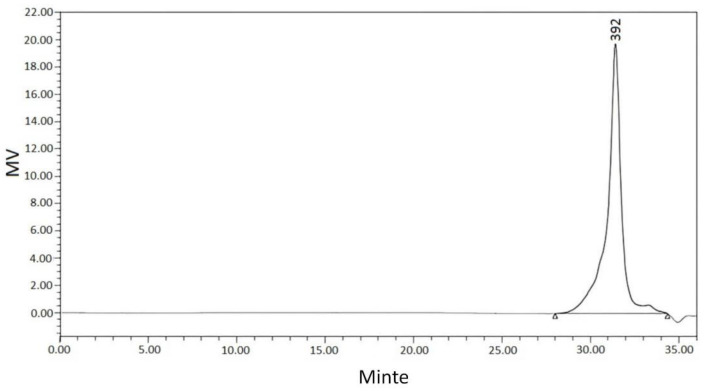
GPC of benzoxazine (bisphenol A).

**Figure 13 materials-15-08364-f013:**
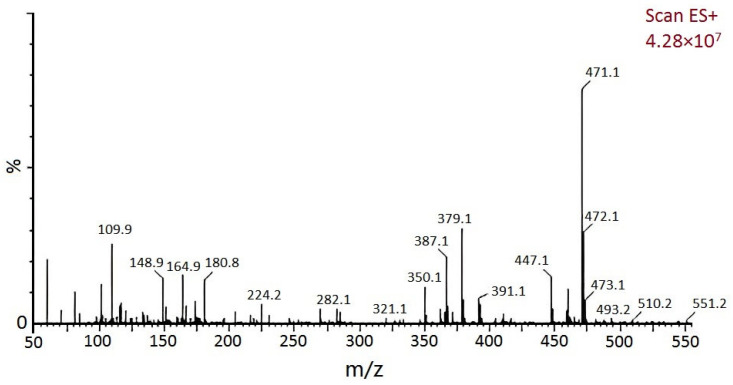
Mass spectrogram of benzoxazine (bisphenol A).

**Figure 14 materials-15-08364-f014:**
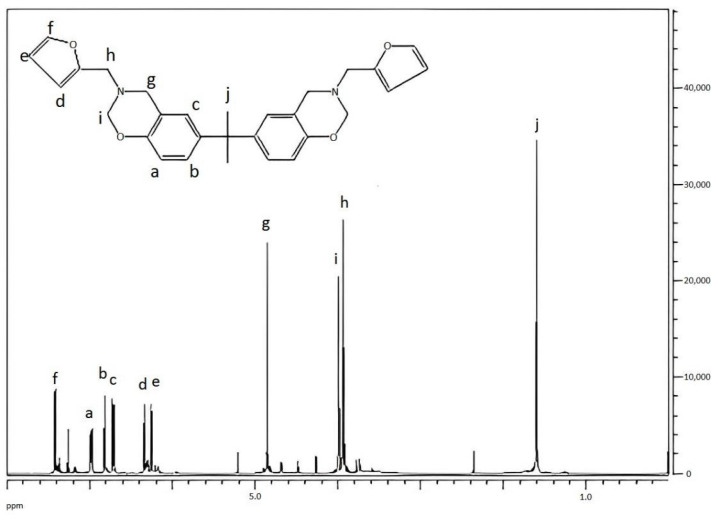
^1^H-NMR spectrum of benzoxazine (bisphenol A).

**Figure 15 materials-15-08364-f015:**
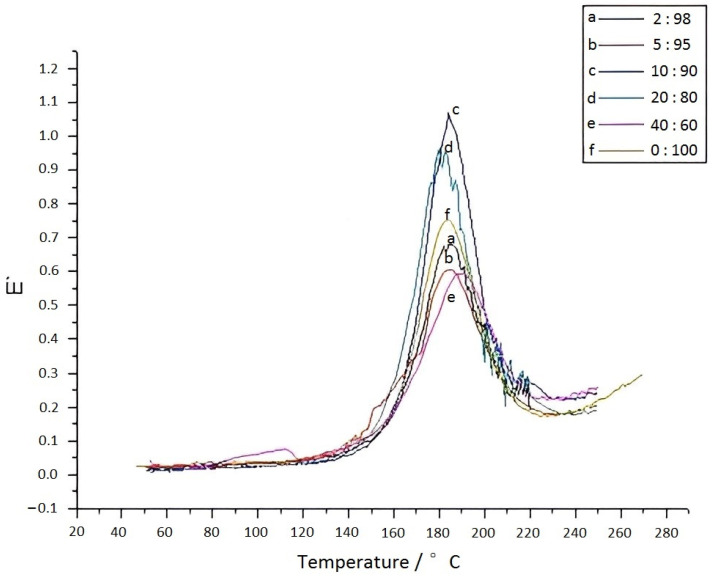
Plots of temperature versus loss factor for BZ composite material with different eugenol–furfuryl amine contents.

**Figure 16 materials-15-08364-f016:**
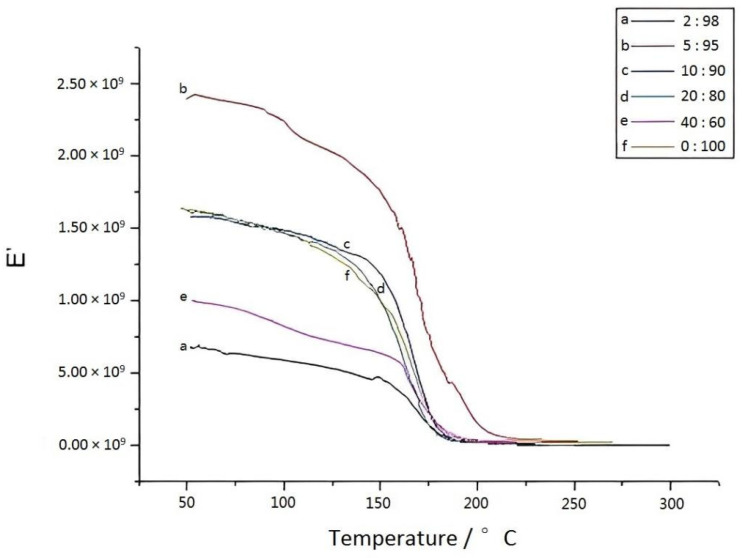
Plots of temperature versus storage modulus for BZ composite material with different eugenol–furfuryl amine content.

**Figure 17 materials-15-08364-f017:**
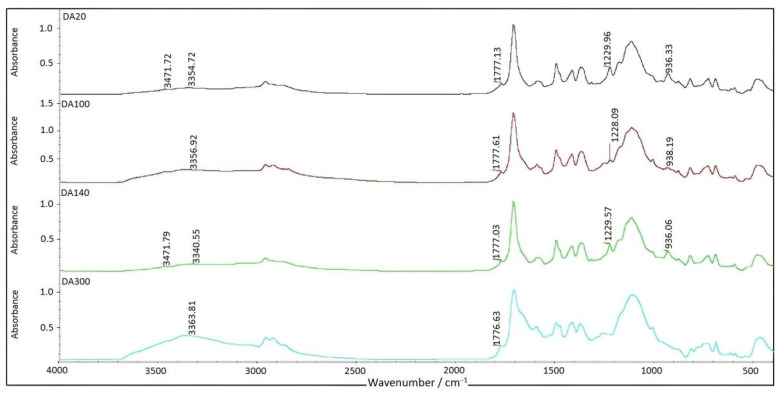
FTIR spectrum of OMPS/bisphenol A-furfuryl amine oxazine (1:1).

**Figure 18 materials-15-08364-f018:**
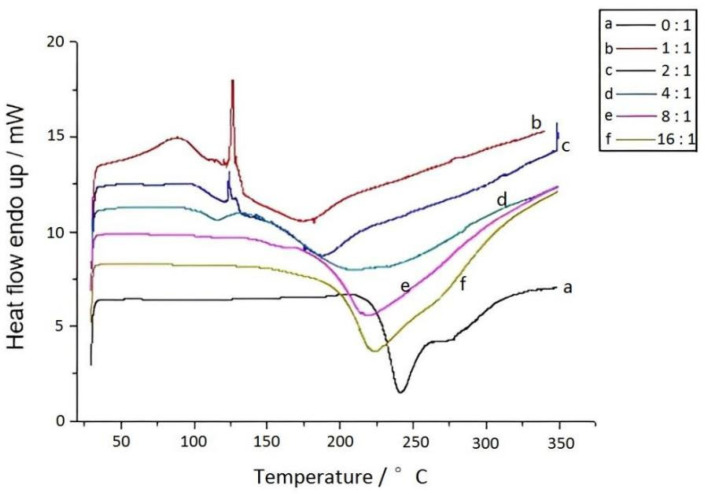
DSC curve of benzoxazine (cardanol) with different content of OMPS.

**Table 1 materials-15-08364-t001:** The product yield with different solvents.

Solvent	Dielectric Constant (20 °C)	Yield	Product Status
Toluene	2.240	52.81%	White powder
Chloroform	4.900	————	Yellow crystalline solid
n-Butanol	17.84	————	————
DMF	36.71	————	White flocculent solid

Note: Eugenol (0.1 mol): 16.4 g; furfurylamine: (0.1 mol) 9.7 g; and paraformaldehyde: (0.2 mol) 6 g.

**Table 2 materials-15-08364-t002:** The product yield with different aldehydes.

Species	Theoretical Yield/g	Actual Yield/g	Yield
Paraformaldehyde	28.5	15.05	52.81%
Trioxane	28.5	————	————
Formaldehyde	28.5	10.33	36.25%

Note: Eugenol (0.1 mol): 16.4 g; furfurylamine (0.1 mol): 9.7 g; and toluene: 100 mL.

**Table 3 materials-15-08364-t003:** The product yield with the quantity of aldehydes.

Paraformaldehyde Dosage	Theoretical Yield/g	Actual Yield/g	Yield
0.2 mol	28.5	15.05	52.81%
0.3 mol	28.5	18.18	63.79%
0.4 mol	28.5	22.08	77.65%
0.5 mol	28.5	22.13	77.47%
0.6 mol	28.5	21.79	76.46%

Note: Eugenol (0.1 mol): 16.4 g; furfurylamine (0.1 mol): 9.7 g; and toluene: 100 mL.

**Table 4 materials-15-08364-t004:** The result of FTIR spectrum of benzoxazine (eugenol) [15].

Functional Group Peak	Position/cm^−1^
Oxazine ring	921
Antisymmetric stretching vibration of carbon–carbon Double bond on allyl group	3112
CH_3_ anti-symmetric stretching vibration	2914
C=C double-bond stretching vibration	1638
Skeleton vibration of benzene ring	1590/1489
Symmetrical deformation of methyl group	1353
Rocking peak of methylene in oxazine ring	1228

**Table 5 materials-15-08364-t005:** The result of FTIR spectrum of benzoxazine (bisphenol A) [15,16].

Functional Group Peak	Position/cm^−1^
Oxazine ring	935
Hydrocarbon anti-symmetric stretching vibration on the carbon–carbon double bond of allyl	3117
CH_3_ anti-symmetric stretching vibration	2965
C=C double bond stretching vibration	1612
Skeleton vibration of benzene ring	1499
Symmetrical deformation of methyl group	1361
Rocking peak of methylene in oxazine ring	1230

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
