# Peer review of "Synthesis and Characterization of Benzoxazine Resin Based on Furfurylamine"

_materials, 2022, doi:10.3390/ma15238364_

Round 1
Reviewer 1 Report
The manuscript “Synthesis and Characterization of Benzoxazine Resin Based on Furfurylamine” by Jing Wang, Jiangnan Yu, Riwei Xu, Chengzhong Wang and Jinping Xiong are not recommended to be published in this journal.
After reviewing the manuscript, I have to inform you that I cannot consider your manuscript in its current form for publication, and thus I must REJECT this manuscript. There are major issues the author needs to fixed in the manuscript.
Abstract:
1. The authors should add the quantitative data (review data) in the abstract section.
2. DSC is not used to confirm the synthesis of target compounds but to study the thermal behaviour of the compounds. Please revised the statement in lines 12-13.
Introduction: The authors also need to discuss deeply on the background of study and the problem that the authors need to overcome.
Experiment:
1. The solvent used for synthesis/reaction is not stated in the methodology.
2. Statement: Place a certain amount of the above blended system in a mold and cure it in a vacuum oven (lines 97-98) à Please revise the statement (change to past tense)
3. Statement: Testing 117 by nuclear magnetic resonance (Bruker Avance Model 600 MHZ NMR with CDCl3 as a solvent, and Si(CH3)4 as an internal standard at room temperature). à Si(CH3)4 as an internal standard? what do you mean by internal standard at room temperature? Need to mention it?
4. The next statement: In N2 atmosphere, 119 the heating rate was 10°C/min. The reference temperature was Al2O3 à The testing refer to what type of characterization? NMR or DSC? Please clarify the methods.
Results:
1. In section 3.1.1, different solvent used was stated with different percentage yields, however no yield was stated for chloroform, butanol and DMF. In contrast, the yields stated in this Table 3-1 is not same as mentioned in Section 2.1.1. In this section, the authors mentioned that the yield was 77.47% which contradict with the percentage yields in section 3.1.1.
2. What is the propose of dielectric study in Section 3.1.1? For optimization? For my opinion, this is not compulsory in this study and the authors should focussed on the characterization methods to confirm the product.
3. What is the propose of Table 3-2 and Table 3-3 in this research? This data is not required for the synthesis work.
4. Please report the FTIR results in Table form (Sections 3.2 & 3.3)
5. Please remove the uncomplimentary peak in FTIR spectrum (Fig 3-2 & Fig 3-6)
6. According to the figure, Benzoxazine (Eugenol) had a molecular weight of 286, which was consistent with its theoretical molecular weight of 285.1. (Lines 174-175) à Please mention no.of figure in this statement. Any references or previous study to confirm this value?
7. As a result, it was presumed that monomeric cures were not 185 possible for the subclass of pure oxazine à Any reason for this behaviour?
8. In section 3.4, the authors mentioned about DMA testing but in the not stated in the abstract or methodology section. Please revise all the instrument used.
9. There was evidence that a reversible reaction occurred between 20 and 100 degrees Celsius à Please use the right symbols (line 244)
10. In section 3.5, the authors should rearrange the data accordingly. Not from DSC discussion and suddenly come to FTIR discussion. Please be more scientific in reporting the data.
Conclusion: Please revise the instrument used for characterization. The conclusion section should be more specific and conclude the purpose of the study.
References: The reference format is not consistent. Please revise it. Please add more references as some references is missing in the stated statement.
English: The English style should be revised in depth.
Author Response
Dear reviewer,
Thanks so much that you had given these helpful suggestions. Firstly, I have to say sorry for my delay to reply as I have changed my institute and all the following revision and fund resource have to be carried out in the new unit. Please forgive my late response and give me a chance for the publication.
I really appreciate your hard work and these professional ideas! I have given my responses one by one in the attachment and the same changed places were also highlighted in yellow in the revised manuscript.
Thousands of thanks again for your help and please do not hesitate to contact with me if there is any problem.
Sincerely yours,
Jing Wang

Reviewer 2 Report
In the article entitled, “Synthesis and Characterization of Benzoxazine Resin Based on Furfuryl amine “the authors have examined the synthesis of benzoxazines from eugenol and bisphenol A with furfuryl amine, respectively, using natural raw materials.
I think this manuscript can be published in this journal with the following minor revisions.
1) My observation is: Mainly missing all figures and compound structures quality, please improve the resolution of all figures.
2) The author needs to adjust and insert good structure and images.
Author Response

(The authors gave the same response as above.)

Reviewer 3 Report
Comments to the Author:
In this manuscript titled, “Synthesis and Characterization of Benzoxazine Resin Based on Furfurylamine” here the authors claimed the synthesis and investigations of Benzoxazine resin using Bisphenol A, which a high performance, sustainable and eco-efficient material used in a large variety of essential everyday applications in plastic. The idea adopted in this study is somewhat interesting. The manuscript is written in a well way and seems like a story. However, there are few minor things along with some typos, mentioned below must be revised for final draft.
1. In lines 11-13 (abstract), they mentioned that “To confirm the synthesis of the target molecules, IR spectroscopy, GPC, mass spectrometry, H-spectrum NMR and DSC tests were conducted.” DSC determines the temperature and heat flow associated with material transitions as a function of time and temperature, so it is particularly applied to monitor the changes of phase transitions rather than the confirmation of the synthesis of the resin. DSC description in conclusion line 258 may be rewritten as mentioned priory.
2. In lines 76 and 87 (section 2.1), 1mol/L of alkali is needed to be added but the name or type of alkali is missing. As it is, a necessary step to obtain the product so name or type of the alkali used must be mentioned.
3. DSC instrumentation is missing in line 119 (Section 2.2).
4. IR peaked should be replaced by IR peaks in line 162 (Section 3.2).
5. CH3 should be written as CH3 in line 164 (Section 3.2).
6. Figure 3-9 is mistakenly mentioned as Figure 3-10 in line 219 (Section 3.4).
7. Lines 262-265 (section 4), “Additionally, as octamaleimidophenyl POSS was added to Bisphenol A-Furfurylamine type oxazine, the curing temperature decreased; as octamaleimidophenyl POSS was added to Bisphenol A-Furfurylamine type oxazine, the decrease in curing temperature grew”; needed to rephrased in a simple way as it is so confusing to understand.
8. Full form of few of the abbreviations like POSS (line 18), Tg (line 54), DMA (line 218), OMPS (Lines 66, 101, 102, 104, 108, 233, 236, 227 and so many others) used frequently were not mentioned when used for the first time.
9. Fig 3-2. FTIR spectrum of Benzoxazine (Eugenol), Fig.3-3 GPC of Benzoxazine (Eugenol), Fig.3-5 DSC Curve of Benzoxazine (Eugenol), Fig.3-6 FTIR Spectrum of Benzoxazine (Bisphenol A), and Fig.3-7 GPC of Benzoxazine (Bisphenol A) can be drawn using excel or Origin software for better presentations rather than their scanned images.
10. Image quality and resolution of Fig.3-9 Plots of Temperature versus Loss Factor for BZ Composite Material with different eugenol- furfuryl amine Content, Fig.3-10 Plots of Temperature versus Storage Modulus for BZ Composite Material with different Eugenol -Furfuryl Amine Content and Fig.3-12 FTIR Spectrum of OMPS / Bisphenol A-furfuryl amine Oxazine (1:1) can be improved.

Author Response

(The authors gave the same response as above.)

Round 2
Reviewer 1 Report
The authors have been added some of the improvement to the manuscript. However, there are still error and comments need to be improved before acceptance.
1. Please revise the statement: To synthesize new benzoxazines, eugenol and bisphenol 10 A were separately reacted with furfurylamine, with the highest yield of eugenol-furfurylamine ben- 11 zoxazine 77.65% and bisphenol A-furfurylamine benzoxazine 93.78% --> Confusing.
2. the characteristic peaks of BZ monomered at 9366 cm-1 and 1229.936 cm-1 disappeared --> 9366 cm-1???? and 1229.936 - please standardize the decimal point.
3. Antisymmetric stretching vibration of carbon carbon --> Antisymmetric stretching vibration of carbon-carbon
4. Please standardize the degree (℃) symbol.
5. Technical Issues: (i) ... shown in figure 3-2 ---> ... shown in Figure 3-2 or followed the style in figure caption (... shown in Fig. 3-2); (ii) 248 ℃ to 175 ℃ --> 248 to 175 ℃; (iii) 9366 cm-1 and 1229.936 cm-1 --> 9366 and 1229.936 cm-1
6. room temperature-350℃ --> room temperature to 350℃
7. H-spectrum NMR --> 1H-NMR
8. Please provide the 1H-NMR spectral discussion with the evident of the spectrum in the manuscript.
9. Please remove the unnecessary peaks in Fig.3-11.
10. The author mentioned that "There was previous study in our laboratory, which had been applied for a patent, Patent No.: CN201010178513.3." However this patent was not cited in this study. Please cite it in the text and manuscript.
11. certain amount of solvent tetrahydrofuran --> certain amount of tetrahydrofuran (please check the other statement)
12. The English style still need to be revised in depth.
Author Response
Dear reviewer,
Thousands of thanks for these helpful suggestions.
I really appreciate your hard work and these professional ideas! I have given my responses one by one in the attachment and the same changed places were also highlighted in blue in the revised manuscript.
Thanks again for these useful suggestions and please do not hesitate to contact with me if there is any problem.
Sincerely yours,
Jing Wang

Round 3
Reviewer 1 Report
All the comments have been amended and revised. Thus, I agree with this manuscript to be published.
Thank you. Regards.